# Exploring optimal granularity for extractive summarization of unstructured health records: Analysis of the largest multi-institutional archive of health records in Japan

Kenichiro Ando[1,2,3], Takashi Okumura[4]*, Mamoru Komachi[1], Hiromasa Horiguchi[3], Yuji Matsumoto[2]

1 Graduate School of Systems Design, Tokyo Metropolitan University, Tokyo, Japan, 2 Center for Advanced Intelligence Project, RIKEN, Tokyo, Japan, 3 National Hospital Organization, Tokyo, Japan, 4 School of Regional Innovation and Social Design Engineering, Kitami Institute of Technology, Hokkaido, Japan

* tokumura@mail.kitami-it.ac.jp

**Data Availability Statement:** The NHO data we used in this study is a subset of NCDA (NHO Clinical Data Archives), which is an archive of

## Abstract

Automated summarization of clinical texts can reduce the burden of medical professionals. "Discharge summaries" are one promising application of the summarization, because they can be generated from daily inpatient records. Our preliminary experiment suggests that 20–31% of the descriptions in discharge summaries overlap with the content of the inpatient records. However, it remains unclear how the summaries should be generated from the unstructured source. To decompose the physician's summarization process, this study aimed to identify the optimal granularity in summarization. We first defined three types of summarization units with different granularities to compare the performance of the discharge summary generation: whole sentences, clinical segments, and clauses. We defined clinical segments in this study, aiming to express the smallest medically meaningful concepts. To obtain the clinical segments, it was necessary to automatically split the texts in the first stage of the pipeline. Accordingly, we compared rule-based methods and a machine learning method, and the latter outperformed the formers with an F1 score of 0.846 in the splitting task. Next, we experimentally measured the accuracy of extractive summarization using the three types of units, based on the ROUGE-1 metric, on a multi-institutional national archive of health records in Japan. The measured accuracies of extractive summarization using whole sentences, clinical segments, and clauses were 31.91, 36.15, and 25.18, respectively. We found that the clinical segments yielded higher accuracy than sentences and clauses. This result indicates that summarization of inpatient records demands finer granularity than sentence-oriented processing. Although we used only Japanese health records, it can be interpreted as follows: physicians extract "concepts of medical significance" from patient records and recombine them in new contexts when summarizing chronological clinical records, rather than simply copying and pasting topic sentences. This observation suggests that a discharge summary is created by higher-order information processing over concepts on sub-sentence level, which may guide future research in this field.

actual patient health records replicated from national hospitals throughout Japan. Because of the nature of the data, access to the dataset is strictly restricted to research approved by the Ethics Review Board of the National Hospital Organization, Japan, and cannot be released publicly. However, once IRB approves, researchers would be able to obtain the dataset in the same manner as we did for the study. The overview and the contact address of the archive is available below. https://nho.hosp.go.jp/cnt1-1_000070.html.

**Funding:** The work was funded by Center for Advanced Intelligence Project, Riken, Japan. The funder had no role in study design, data collection and analysis, decision to publish, or preparation of the manuscript.

**Competing interests:** The authors have declared that no competing interests exist.

## Author summary

Medical practice includes significant paperwork, and therefore, automated processing of clinical texts can reduce medical professionals' burden. Accordingly, we focused on hospitals' discharge summaries from daily inpatient records stored in Electric Health Records. By applying summarization technologies, which are well-studied in Natural Language Processing, discharge summaries could be generated automatically from the source texts. However, automated summarization of daily inpatient records involves various technical topics and challenges, and the generation of discharge summaries is a complex process of mixing extractive and abstractive summarization. Thus, in this study, we explored optimal granularity for extractive summarization, attempting to decompose actual physicians' processing. In the experiments, we used three types of summarization units with different granularities to compare performances of discharge summary generation: whole sentences, clinical segments, and clauses. We originally defined clinical segments, aiming to express the smallest medically meaningful concepts. The result indicated that sub-sentence processing, larger than clauses, improves the quality of the summaries. This finding can guide future development of medical documents' automated summarization.

## 1 Introduction

Automated summarization of clinical texts can reduce the burden of medical professionals because their practice includes significant paperwork. A recent study found that family physicians spent 5.9h in an 11.4h workday on electronic health records (EHRs) [1]. In 2019, 74% of physicians spent more than 10h per week [2]. Another study reported that physicians spent 26.6% of their daily working time on documentation [3].

Compilation of hospital discharge summaries is an onerous task for physicians. Because daily inpatient records are already filed in the systems, computers might efficiently support physicians by generating summaries of clinical records. Although research has been conducted to identify certain classes of clinical information in clinical texts [4–8], there has been limited research on acquiring expressions that can be used to write discharge summaries [9–14]. Because many summarization techniques have been developed in natural language processing (NLP), the generation of discharge summaries can be a promising application of the technology.

However, automated summarization of daily inpatient records involves various technical topics and challenges. For example, descriptions of important findings related to a patient's diagnosis require an extractive summary. Our preliminary experiments revealed that 20–31% of the sentences in discharge summaries were created by copying and pasting. This result proves that a certain amount of content can be automatically generated by extractive summarization. Meanwhile, when a patient is discharged from the hospital after surgery without any major problems, it is necessary to summarize the clinical record as the patient "recovered well after the surgery," even if more details of the postoperative process are described in the records. Therefore, such descriptions cannot be created by copy and paste, and needs to be abstracted. These observations suggest that the generation of discharge summaries is a complex process that is a mixture of extractive and abstractive summarization, and it remains unclear how to process the unstructured source texts, i.e., free-texts. To advance this research field, it is desirable to properly decompose these summarization processes and clarify their interactions.

To this end, this study focuses on the extractive summarization process by physicians. Some recent studies investigated the best granularity units in this type of summarization [15, 16]. However, the granularity of extraction has not been explored for the summarization of medical documents. Thus, we attempted to identify the optimal granularity in this context, by defining three units with different granularities and comparing their summarization performance: whole sentences, *clinical segments*, and clauses. The *clinical segments* are our novel concepts to express the smallest medically meaningful concepts and are detailed in the methodology section (Section 3).

This paper is organized as follows. In Section 2, we survey related work. Section 3 describes the materials and methods. Section 4 presents the experiment and its results, and Section 5 discusses the experiment. Finally, Section 6 concludes the paper.

## 2 Related work

Automated summarization is an actively studied field [15–19] with two main approaches: extractive and abstractive summarization. The former extracts contents from source texts, whereas the latter creates new contents. Generally, the abstractive approach provides more flexibility in summarization but often produces fake contents that do not match the reference summary, which is referred to as "hallucination" [19–21]. Thus, in the medical field, "extractive summarization" has been mainly used for knowledge acquisition of clinical features such as diseases, prescriptions, examinations, etc. The determination of the optimal granularity would lead to the more reliable information. Secondly, the precise spanning of extraction would read to avoid extraction of unnecessary information, keeping the precision of the processing high.

Meanwhile, Natural Language Processing on unstructured medical text has been focusing on normalization and prediction, such as ICD codes, mortality, or readmission risk [22–27]. However, they are not summarization in a narrow sense, that distills important information from the input. Several works targeted acquiring key information such as disease, examination result, or medication from EHRs [6, 8, 28, 29], while these studies collected fragmented information and did not try to generate contextualized passage. There are a line of researches that targeted to help physicians get the point quickly by generating a few key sentences [7, 30–32]. However, most studies that producing discharge summaries used structured data as input. [33–35]. Some other studies attempted to generate discharge summaries from free-form inpatient records, as we anticipated [9–14]. In part, an encoder-decoder model was used to generate sentences for abstractive summarization [9–11]. These studies can create a whole document of discharge summary. However, this approach may result in hallucinations, which limits its clinical use, although data can be corrected manually by physicians before filing. The other studies summarized sentences, using extractive summarization [11–14], and unsupervised generation using prompt engineering [36, 37] would further contribute to the performance, although they can not generate entire texts.

For advancing the research on summarization of clinical texts, appropriate language resources are indispensable. In English, public corpora of medical records are available, such as MIMIC-III [38, 39], and [40]. However, the number of resources available in Japanese is highly limited. The largest publicly available corpus is the one used for a shared task in an international conference, NTCIR [41]. A non-profit organization for language resources maintains another corpus, GSK2012-D [42]. However, their data volume is small, and their statistics exhibit significant difference from those of large-scale data, as illustrated in Table 1. This low-resource situation makes the processing of Japanese medical documents more challenging. First, Japanese medical texts often contain excessive shortening of sentences and

**Table 1. Statistics of the target data.**

| Inpatient records | | | | |
|---|---|---|---|---|
| Dataset | Cases | Sentences/Document | Words/Sentence | Characters/Sentence |
| NHO data | 24,641 | 192.0 | 9.0 | 18.1 |
| GSK2012-D | 45 | 97.4 | 7.5 | 15.1 |
| MedNLP | 278 | 22.6 | 12.7 | 22.4 |
| Our corpus | 108 | 274.1 | 9.1 | 18.5 |
| Discharge summary | | | | |
| Dataset | Cases | Sentences/Document | Words/Sentence | Characters/Sentence |
| NHO data | 24,641 | 35.0 | 12.4 | 23.3 |
| Our corpus | 108 | 17.4 | 18.6 | 34.4 |

orthographical variants of terms originating from foreign languages. Besides, Japanese requires word segmentation. Most importantly, there is no Japanese parallel corpus of inpatient records and discharge summaries. Therefore, we built a new corpus as detailed in the next section.

## 3 Materials, method, and preprocessing

### 3.1 Target text

Clinical records can be expressed in various dialects and jargons. Accordingly, a study on a single institution would lead to highly biased results in medical NLP tasks because of local and hospital-specific dialects. To explore the optimal granularity for clinical document summarization, it is necessary to conduct a multi-institutional study to mitigate the potential bias caused by the medical records stored in a single EHR source. For this purpose, we designed an experiment using the largest multi-institutional health records archive in Japan, National Hospital Organization Clinical Data Archives (NCDA) [43]. NCDA is a data archive operated by the National Hospital Organization (NHO), which stores replicated EHR data for 66 national hospitals owned by this organization. Thus, the archive has become a valuable data source for multi-institutional studies that span across the country.

On this research infrastructure, informed consent and patient privacy are ensured in the following manner. At the national hospitals, notices about their policy and the EHR data usage are posted in their facilities. The patients who disagree with the policies are supposed to notify the hospital by an opt-out form, to be excluded from the archive. Likewise, minors and their parents can turn in the opt-out form, at will. To conduct a study on the archive, researchers must submit their research proposals to the institutional review board. Once the study is approved, the data are extracted from NCDA, and anonymized to construct a dataset for further analysis. The data are accessible only in a secured room at the NHO headquarters, and only statistics are allowed to be carried out of the secured room, for protection of patients' privacy.

In this present research, the analysis was conducted under the IRB approval (IRB Approval No.: Wako3 2019-22) of the Institute of Physical and Chemical Research (RIKEN), Japan, which has a collaboration agreement with the National Hospital Organization. The dataset we used for the study, referred to as **NHO data** hereafter, is the anonymized subset of the archive, which includes 24,641 cases collected from five hospitals that belong to the NHO. Each case includes inpatient records and a discharge summary for patients of internal medicine departments. The statistics of the target data are summarized in Table 1. As shown, the scale of the NHO data is much larger than that of GSK2012-D and MedNLP, which have been used in

previous studies [41]. Accordingly, the results obtained using the NHO dataset are expected to be more general.

## 3.2 Design of the analysis

To identify the optimal granularity of extractive summarization, there are two approaches. One approach is a method that takes *n* word sequences of arbitrary lengths and compares them as the units for summarization. The other approach is a method that uses predefined linguistic units. Previous studies in this domain have used the latter approach and found that a sentence was a longer-than-optimal granularity unit for extractive summarization, as mentioned in Section 1. Another study adopted a clause as a shorter self-contained linguistic unit [44] instead of a sentence [15]. However, it remains unclear whether the clause performs the best in the summarization of clinical records or there could be further possibilities. In this study, we adopt both of the two methods. However, the examination using linguistic units in Japanese is a little different from that in English. In particular, clauses in Japanese have significantly different characteristics from clauses in English because they can be formed by simply adding a particle to a noun. Owing to this characteristics, Japanese clauses are often very short at the phrase level. Accordingly, they cannot constitute a meaningful unit that carries concepts of medical significance. Therefore, we need a self-contained linguistic unit that has a longer span than a clause in Japanese and expresses the smallest medically meaningful concept.

For this reason, we defined the *clinical segment* that spans several clauses but is shorter than a sentence. As exemplified in Table 2, segments may comprise clauses connected by a conjunction to form a medically meaningful unit; alternatively, they may be identical to clauses. For the statistical analysis, the clinical segment must be defined formally so that a splitter can automatically divide sentences into segments. We also need a corpus to train the splitter and evaluate its performance.

When designing the clinical segment, we attempted to extract the atomic events related to medical care as a single unit. For example, statements such as "jaundice was observed in the patient's conjunctiva," "the patient was diagnosed with hepatitis," and "a CT scan was performed" would lose their medical meaning if they are further split. In addition, medical events are the central statements in medical documents, whereas non-medical events play a relatively small role. Therefore, in this study, we considered only medical events as a component of self-contained units, and non-medical events were interpreted as noise. In previous studies, a self-contained unit was defined with respect to semantics. In our study, it was extended to a pragmatic unit based on domain knowledge. The details of the six segmentation rules are listed in Table 3.

Based on this definition, we built a small corpus for the segmentation task. We used an independent dataset that included inpatient records and their discharge summaries for 108 cases. This corpus was built because annotation over the NHO data was restricted due to privacy concerns. The statistics of the resulting corpus are given in Table 1 (**Our corpus**). With respect to the inpatient records, the corpus is closer to real data than in previous studies, except for the number of sentences in a document. For the discharge summary, there are no publicly available Japanese corpora besides the one we built. Because of the summarization process, the sentences contain more words and characters than the source inpatient records. The total number of segments in the corpus was 3,816, the average number of segments per sentence was 2.18, and the average number of segment boundaries per sentence was 1.18. The agreement rate between the participants of the segmentation task and an author is 0.82, which is sufficiently high to be used for further study. The agreement rate is the accuracy of the workers'

**Table 2. Examples of the three types of units.**

| Units | Examples |
|---|---|
| Sentence | 認知症が進んでおり自宅退院は困難であること、施設入居のためにはご家族の手続きが必要になることを説明 <br><br> (We explained that it would be difficult to discharge her due to her advanced dementia, and that her family would need to make arrangements to move her into another facility.) |
| Segment | 認知症が進んでおり **SEP** 自宅退院は困難であること、**SEP** 施設入居のためにはご家族の手続きが必要になること を **SEP** 説明 <br><br> (Due to her advanced dementia **SEP** it would be difficult to discharge **SEP** her family would need to make arrangements to move her into another facility **SEP** we explained) |
| Clause | 認知症が進んでおり **SEP** 自宅退院は **SEP** 困難である **SEP** こと、**SEP** 施設入居のためには **SEP** ご家族の手続きが必要になることを **SEP** 説明 <br><br> (Due to her advanced dementia **SEP** discharge **SEP** it would be difficult **SEP** (verb nominalizer) **SEP** to move her into another facility **SEP** her family would need to make arrangements **SEP** we explained) |
| **SEP** | **SEP** indicates the boundary of either a segment or clause. |

**Table 3. Segmentation rules.**

**Rule 1** *Split at the end position of a predicate, by a comma or a verbal noun.*
This is the base rule for segmentation, and others are exception rules.
(e.g., "絶食、 **SEP** 抗菌薬投与で **SEP** 肺炎は軽快。")
(e.g., "(After) fasting and SEP antibiotic use, SEP pneumonia was relieved.")

**Rule 2** *If a segment is enclosed in parentheses, split a sentence at the positions of parentheses.*
To extract the clinical segment inside parentheses, parentheses sometimes become segment boundaries.
(e.g., "画像で「 SEP 両側肺門部に陰影あり、 SEP CT で両肺に多彩な浸潤影を認め SEP 重症肺炎」 SEP として4月10日に入院。")
(e.g., "On imaging, "SEP there are bilateral hilar shadows and SEP widespread consolidation in both lungs on CT scan, SEP (suspected of) severe pneumonia" SEP (the patient was) admitted to the hospital on April 10.")

**Rule 3** *Split content that includes disease name.*
Disease names are often written as diagnoses and play an important role in EHRs. Therefore, even if rule 1 does not match, the content that includes disease names should be split.
(e.g., "肺炎疑いで SEP 当院紹介となった。")
(e.g., "Due to suspected pneumonia, SEP he was referred to our hospital.")

**Rule 4** *Split examination results and their evaluation.*
Examination results and their evaluation are often written in a single sentence. Because the meaning of the examination results and their evaluation are clearly different, they should be divided even if rule 1 does not match.
(e.g., "血清クレアチニンキナーゼは4512 U/L と 4512 U/L と SEP 高度に上昇していた。")
(e.g., "Serum creatinine kinase level was 4512 U/L, SEP which was highly elevated.")

**Rule 5** *Do not split anything that is not related to the medical treatment.*
If the content is medically meaningless, its role is not important in its document, and it is not worthy of analysis. Therefore, the content with little relevance to medical treatment is not split, even if it matches rule 1.
(e.g., "ケアマネジャーに同伴されて来院した。")
(e.g., "She came to our hospital accompanied by her care manager.")

**Rule 6** *Do not split content that does not add meaning.*
If the content that supplements the meaning of the previous description does not add meaning (e.g., "...schedule to [VP] ..." and "...continue the treatment ..."), it is not split even if it matches rule 1.
(e.g., "外来で抜糸を行う方針とした。")
(e.g., "It was planned to remove sutures as an outpatient.")
This includes contents where the semantic label does not change before and after the split.
(e.g., "発熱、 盗汗、 体重減少、 喀痰、 血痰は否定。")
(e.g., "Fever, sweating, weight loss, sputum, and bloody sputum were not observed.")
It also includes contents that represent the passage of time or assumptions.
(e.g., "抗菌薬開始後、 発熱 腹痛は徐々に改善し")
(e.g., "After starting antibiotic use, fever and abdominal pain gradually improved.")

labels for the correct boundaries annotated by an author. Across this task, we adopted the labels annotated by one of the authors.

## 3.3 Preprocessing

Table 4 shows a discharge summary—a type of medical record written by a Japanese physician. As illustrated, it is a noisy document: punctuation marks are missing, and line breaks appear in the middle of a sentence. Sentence boundaries may be denoted by spaces instead of punctuation marks. Therefore, for the further analysis of the three types of extraction units, we first need preprocessing for *sentence splitting* and *segment splitting*, which are shown in the upper part of Fig 1.

For sentence splitting, we adopt two naive rules below to define the boundaries of a sentence:

1. A statement that ends with a full-stop mark.

2. A statement that ends with a new line and has no full-stop mark.

**Table 4. Example of a discharge summary.**

| | |
|---|---|
| #1 細菌性髄膜炎 | #1 Bacterial meningitis |
| 4/20　5/8 VCM 1250mg(q12h) | 4/20-5/8 VCM 1250mg (q12h) |
| 4/20 SBT/ABPC 1.5g単回 | 4/20 SBT/ABPC 1.5g single dose |
| 4/20　MEPM 2g(q8h) | 4/20- MEPM 2g (q8h) |
| 4/20　4/23 デキサート 6.6mg(q6h) | 4/20-4/23 Dexate 6.6mg (q6h) |
| 4/20　4/22 日赤ポログロビン | 4/20-4/22 Nisseki polyglobin |
| 4/20 腰椎穿刺1回目髄液糖定量 30 mg/dl(血中糖 95mg/dl) 細胞数 2475/µl. | 4/20 1st lumbar puncture, cerebrospinal fluid glucose level 30 mg/dl (blood glucose level 95 mg/dl), cell count 2475/µl. |
| グラム染色するも明らかな菌が見つからず、髄液培養でも優位な菌は培養されなかった。 | Gram stain did not reveal any obvious bacteria, and cerebrospinal fluid culture also did not reveal any predominant bacteria. |
| 細菌性髄膜炎に対するグラム染色の感度は60%程度であり、培養に関しても感度は高くない。 | The sensitivity of the gram stain for bacterial meningitis is about 60%, and the sensitivity of the culture is not high either. |
| また髄液中の糖はもう少し減るのではないだろうか。 | Also, the glucose in the cerebrospinal fluid would have been slightly lower. |
| 確定診断はつかないものの、最も疑わしい疾患であった。 | Although no definitive diagnosis could be made, bacterial meningitis was the most suspicious disease. |
| 起因菌はMRSA, 腸内細菌等を広域にカバーするためバンコマイシン, メロペネム(髄膜炎dose)とした。 | The causative organism was assumed to be MRSA, and vancomycin and meropenem (meningitis dose) were used to cover a wide range of enteric bacteria. |

The left column shows the original Japanese texts, and the right column shows corresponding English translations.

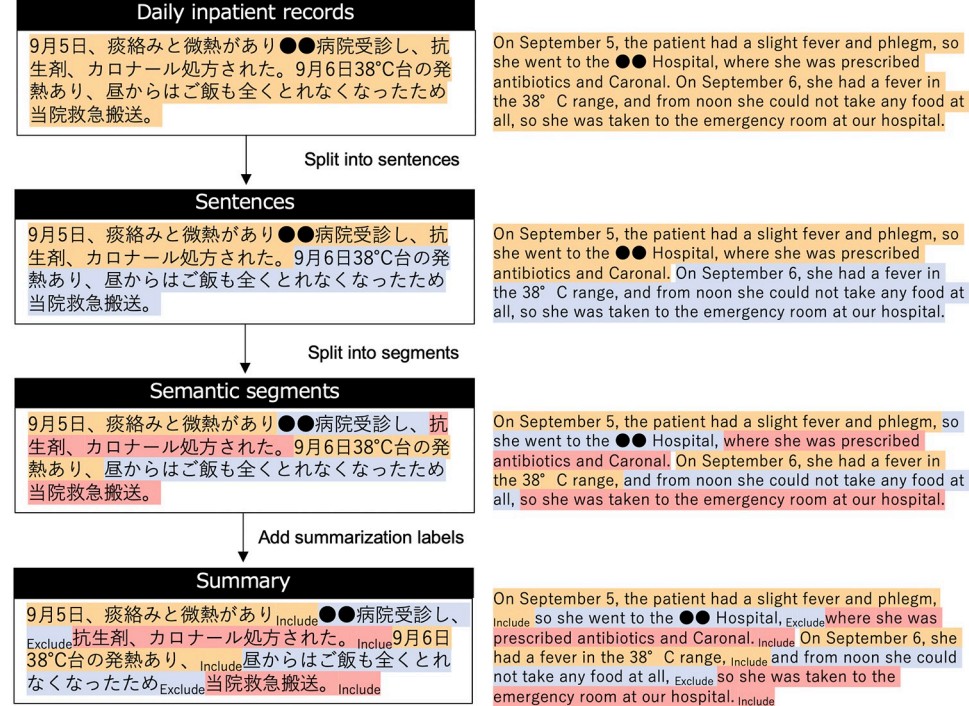

**Fig 1. Outline of our pipeline.** The top block is an example of the inpatient record, and the subsequent blocks indicate the chain of processes up to adding summarization labels.

There is oversimplification here, compared to sentence splitting tasks in medical NLP that have been studied [45, 46]. However, since it is not a focus of this study, we adopted this naive approach for its simplicity. In this process, we also used MeCab [47] as a tokenizer. The MeCab's dictionaries are mecab-ipadic-NEologd [48] and J-MeDic [49] (MANBYO 201905).

Next, sentences must be automatically split into clinical segments to efficiently analyze the huge dataset, NHO data. We compared several approaches to achieve the best splitting performance. In this study, we used 3,816 annotated segments in the corpus and applied six-fold cross-validation.

We used three rule-based splitters as baselines: a simple rule-based model for splitting by full-stop marks (**Full-stop**), another simple rule-based model for splitting by full-stop marks and verbs (**Full-stop & Verb**), and a complex rule-based model for splitting by clauses (**CBAP**) [50]. To be more precise, in the case of the Full-stop & Verb model, it starts with a verb and splits in front of the next occurring noun except for non-independents. The last model, which included 332 rules that were manually set up based on morphemes, was used to confirm that clinical segments have different boundaries than traditional clauses.

We used **SEGBOT** [51] as a machine learning method based on a pointer network architecture [52] for the splitting task. The method includes three phases: encoding, decoding, and pointing. An overview is shown in Fig 2. Medical records may include local dialects and technical terms that are not listed on public language resources. Accordingly, the splitter must handle even unknown words. In our approach, each input word is first represented by a distributed representation using fastText [53, 54]. FastText is a model that acquires vector representations of words considering the context. Notably, fastText can obtain vectors of unknown words by decomposing them into character n-grams. These vectors capture hidden information about a language, such as word analogies and semantics.

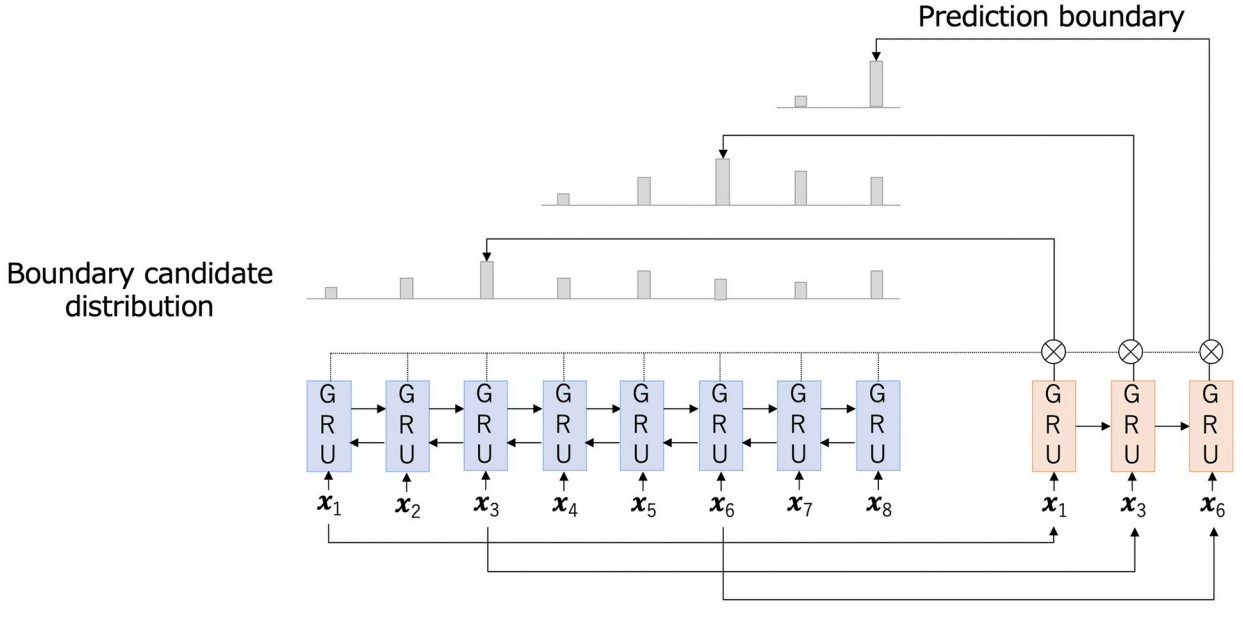

**Fig 2. Overview of SEGBOT.**

**Table 5. Results of the segmentation task.**

|  | Precision | Recall | F1 score |
|---|---|---|---|
| Full-stop | 0.521 | 0.187 | 0.275 |
| Full-stop & Verb | 0.569 | 0.610 | 0.589 |
| CBAP [50] | 0.368 | 0.464 | 0.411 |
| SEGBOT [51] | **0.864** | **0.829** | **0.846** |

The numbers in bold indicate the best performing methods.

The performance of the splitter methods is summarized in Table 5. The machine-learning-based SEGBOT outperformed the others, with its F1 score being 0.257 points higher than that of the Full-stop & Verb model, which was the second best. Since this precision of 0.864 is higher than the inter-annotator agreement, it is considered to be almost the upper bound. In addition, CBAP, which is a clause segmentation model, has a low F1 score of 0.411, suggesting that the definitions of the clause and the clinical segment are inherently different. The precision of the model with splitting at the full-stop marks (Full-stop) is only 0.521, indicating that the clinical segment is not always split at the full-stop marks, and that it is necessary to consider the context for splitting. Overall, the results suggest that machine learning is the best fit for the segmentation task. Thus, the data preprocessed by this method are used for the main experiment of this study.

# 4 Main experiment

In this section, we describe our experimental settings and results of automatic summarization. First, we present the performance metric of the experiments; specifically, the ROUGE score is used as a quality measure for a summary. Next, we describe a summarization model used in the experiments, followed by the datasets used to train the model. Finally, we present the experiments and their results.

## 4.1 Evaluation metric

Measurement of the summarization quality must be automated to avoid costly manual evaluation. ROUGE [55] has been used as a standardized metric to measure the summarization quality in NLP tasks. Formally, ROUGE-N is an n-gram recall between a candidate summary and the reference summaries. When we have only one reference document, ROUGE-N is computed as follows:

$$
\text{ROUGE-N} \quad = \quad \frac{\sum_{gram_n \in Reference} Count_{match}(gram_n)}{\sum_{gram_n \in Reference} Count(gram_n)}, \tag{1}
$$

where $Count_{match}(gram_n)$ is the maximum number of n-grams that co-occur in a candidate summary and a reference summary.

When we have several references, ROUGE-L is the longest common subsequence (LCS) score between a candidate summary and the reference summaries. As it can assess word relationships, it is generally considered a more context-aware evaluation measure than

ROUGE-N. Specifically, ROUGE-L is computed as follows:

$$Recall_{lcs} = \frac{\sum_{i=1}^{u} LCS_{\cup}(\boldsymbol{r}_i, \boldsymbol{C})Reference_{tokens}}{,} \tag{2}$$

$$Precision_{lcs} = \frac{\sum_{i=1}^{u} LCS_{\cup}(\boldsymbol{r}_i, \boldsymbol{C})Summary_{tokens}}{,} \tag{3}$$

$$ROUGE-L = \frac{2Recall_{lcs}Precision_{lcs}}{Recall_{lcs} + Precision_{lcs}}, \tag{4}$$

where $u$ is the number of reference sentences, and $LCS_{\cup}(\boldsymbol{r}_i, \boldsymbol{C})$ is the LCS score of the union of the longest common subsequences between the reference sentence $\boldsymbol{r}_i$ and $\boldsymbol{C}$, where $\boldsymbol{C}$ is the sequence of candidate summary sentences. For example, if $\boldsymbol{r}_i = (w_1, w_2, w_3, w_4)$, and $\boldsymbol{C}$ contains two sentences: $\boldsymbol{c}_1 = (w_1, w_2, w_6, w_7)$ and $\boldsymbol{c}_2 = (w_1, w_8, w_4, w_9)$, the longest common subsequence of $\boldsymbol{r}_i$ and $\boldsymbol{c}_1$ is $(w_1, w_2)$, and the longest common subsequence of $\boldsymbol{r}_i$ and $\boldsymbol{c}_2$ is $(w_1, w_4)$. The union of the longest common subsequences of $\boldsymbol{r}_i$, $\boldsymbol{c}_1$, and $\boldsymbol{c}_2$ is $(w_1, w_2, w_4)$, and $LCS_{\cup}(\boldsymbol{r}_i, \boldsymbol{C})$. = 3/4.

## 4.2 Summarization model

In an extractive summarization task, the goal is to automatically assign a binary label to each unit of the input to indicate whether this unit should be included in the summary. Therefore, we adopted a single classification model to cover the three types of units.

Following Zhou et al. [15], we used a model based on BERT [56], as shown in Fig 3. BERT is a pretrained neural network, and its parameters are learned from a large number of documents in advance. BERT is known to achieve a good accuracy even with few training samples. Instead of the original work that adopted BERT as an encoder for extractive summarization,

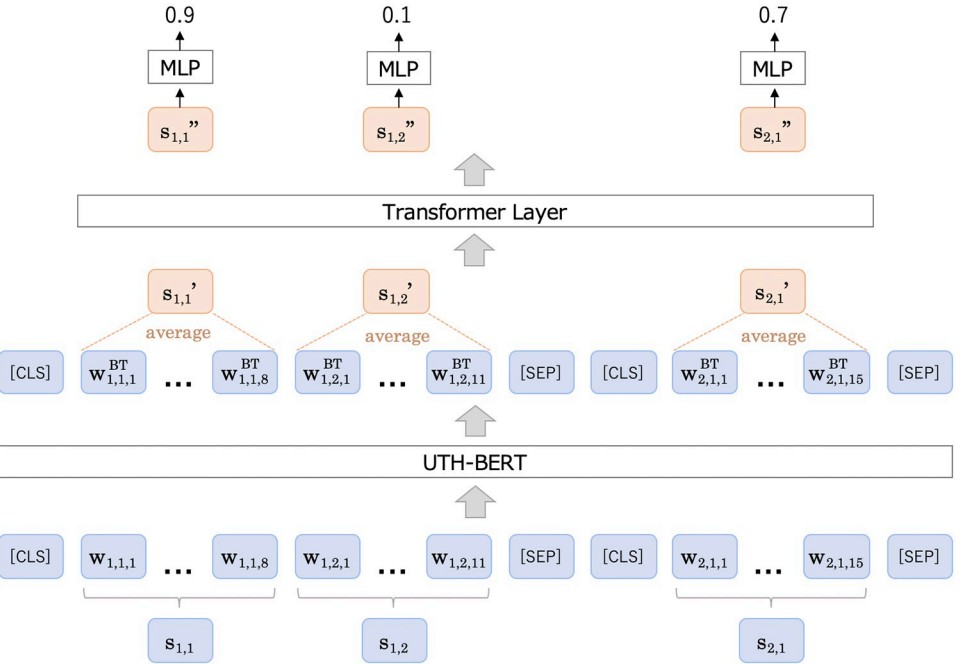

**Fig 3. Overview of classification model for clinical segments.**

we adopted UTH-BERT [57]. In contrast to the previous Japanese BERT models [58–60], which were pre-trained mainly on web data such as Wikipedia, UTH-BERT was pretrained on a large number of Japanese health records and is expected to perform better on documents in the target domain.

Formally, let the $i$-th sentence contain $l$ segments $S_i = (s_{i,1}, s_{i,2}, \ldots, s_{i,l})$. The $j$-th segment with $k$ words in $S_i$ is denoted by $s_{i,j} = (w_{i,j,1}, w_{i,j,2}, \ldots, w_{i,j,k})$. We add [CLS] and [SEP] tokens to the boundaries between sentences. After applying the UTH-BERT encoder, the vector of tokens is represented as $(w^{BT}_{i,j,1}, w^{BT}_{i,j,2}, \ldots, w^{BT}_{i,j,k})$. Next, we apply average pooling at the segment level. The pooled representation $s'_{i,j}$ is formulated as follows:

$$s'_{i,j} \quad = \quad \frac{1}{k}\sum_{1}^{k} w^{BT}_{i,j,k}. \tag{5}$$

Note that segments and clauses do not include the [CLS] and [SEP] tokens in average pooling. Subsequently, we apply a segment-level transformer [61] to capture their relationship for extracting summaries. The model predicts the summarization probability from those outputs as follows:

$$S'' \quad = \quad \text{Transformer}(S'), \tag{6}$$

$$p(s''_{i,j}) \quad = \quad \sigma(W_o s''_{i,j} + b_o), \tag{7}$$

where $S' = (s'_{1,1}, s'_{1,2}, \ldots, s'_{i,j})$ is a sequence of segments input to the transformer, and $S'' = (s''_{1,1}, s''_{1,2}, \ldots, s''_{i,j})$ is a sequence that is the output of the transformer. The training objective of the model is the binary cross-entropy loss given the gold label $y_{i,j}$ and the predicted probability $p(s''_{i,j})$.

This model does not need to change its structure depending on the input units. For clauses, the span of the segments is replaced by that of the clauses. In the case of sentences, the average pooling is not performed; instead, we input the [CLS] token into the transformer.

### 4.3 Training data

Our model requires an entire document for training. However, our corpus could be too small to be used for the training of the model, and would compromise the robustness of the model. Accordingly, we used NHO data as training data by assigning pseudo labels. Following previous studies [15, 16], we used the ROUGE scores to automatically assign gold labels to the three units. We used the ROUGE score both to create the gold labels and to evaluate the model. This may seem unusual, but it is a commonly used approach in previous studies. As ROUGE is correlated with human scores [62], the best summary can be obtained by creating a system that maximizes this score during evaluation, regardless of whether this score was used during training. The labeling steps were as follows.

First, we applied the splitter created in Section 3.3 to the NHO dataset and split it into clauses and clinical segments. In this manner, we easily obtained a larger dataset. We used CBAP as a splitter for clauses and SEGBOT as a splitter for clinical segments.

Second, we measured ROUGE-2 F1 for each unit of the source documents (against the discharge summaries), which were then sorted in descending order of their scores. Thus, we obtained a list of units that were important for our summary.

Third, we selected the units from the topmost part of the list. At this stage, we stopped selecting units when the result exceeded 1,200 characters, which was the average length of the summaries in the NHO data.

**Table 6. Results of the summarization task.**

| Units | ROUGE-1 | ROUGE-2 | ROUGE-L |
|---|---|---|---|
| Sentence | 31.91 | 2.50 | 7.93 |
| Segment | **36.15** | **3.12** | **8.26** |
| Clause | 25.18 | 1.30 | 6.62 |

The numbers in bold indicate the best performing methods.

Finally, we assigned positive labels to the selected units. The entire process yielded the gold standard for the training and evaluation without manual annotation. We randomly selected 1,000 documents each for the development and test sets, and we used the remaining 22,641 documents for the training data.

## 4.4 Experiments and results

In this experiment, we used the three contextual units, instead of the n-gram units, and evaluated their impact on the summarization performance to determine which unit performs the best. The results of summarization, using the three types of units, are shown in Table 6. Comparing the three types of units in granularity, the model with clinical segments scored the highest in ROUGE-1, ROUGE-2, and ROUGE-L. The model with clinical segments outperformed sentences and clauses in summarizing inpatient records.

Table 2 shows that a sentence can contain multiple events and has room for further segmentation. It is certain that sentences are longer than clinical segments and clauses. However, the relation between clinical segments and clauses are unclear. Because ROUGE-1 and ROUGE-2 are measured on the basis of 1-gram and 2-gram, respectively, smaller units are more advantageous in the ROUGE evaluation. Table 7 shows the statistical relation of the three types of units. The first column shows how many units are included in a sentence on average. The second and third columns show the average number of tokens and characters included in each type of units. The result suggests that segments are longer than clauses *on average*. Nevertheless, the difference of a clause and a segment is not significant, at least for the average number of characters. Accordingly, the relationship between clause and clinical segment granularity is worthy of a more detailed analysis.

We ensure the order of the three types of linguistic units, by an additional experiment on word-wise relation between clauses and clinical segments. For any two linguistic units in a sentence, there are four possible relationships (Fig 4): "Equal" is where the two match exactly; "Inclusive" is where a segment completely includes a clause; "Included" is where a clause completely includes a segment; and "Overlap" is where the two overlaps.

We obtained statistics of the four relationships, from all inpatient records and discharge summaries in the NHO data. The results are shown in Table 8. We found that 59.6% of them

**Table 7. Granularity of three units.**

| Units | Units/Sentence | Tokens/Unit | Characters/Unit |
|---|---|---|---|
| Sentence | 1 | 8.98 | 18.06 |
| Segment | 2.18 | 6.42 | 11.83 |
| Clause | **2.75** | **5.74** | **10.74** |

The numbers in bold indicate the smallest units.

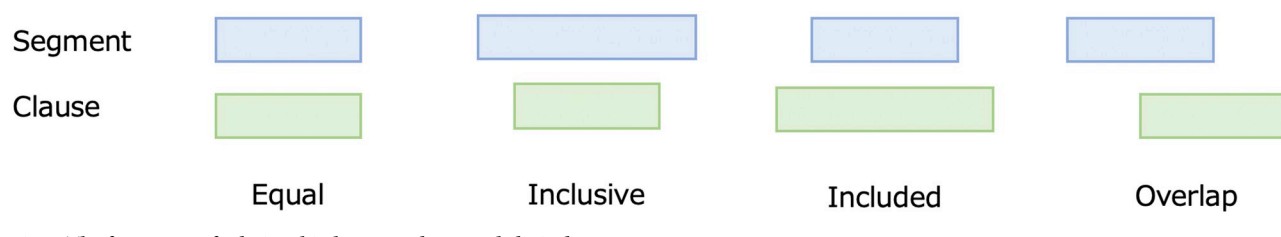

**Fig 4. The four types of relationship between clause and clinical segment.**

have the same boundaries. This is influenced by the many short sentences that have no boundaries. Then, "Inclusive" shared 20.0% of the relations. The sum of "Equal" and "Inclusive" turned out to be 79.6%, which is six times more than "Included" that shared only 13.1%. The figures gives the detailed dynamics of the relation between segments and clauses, shown just as 11.83 and 10.74 characters/unit in Table 7. Although the difference in the average length between segment and clause is small, there is a significant difference between segments and clauses in their relative sizes, when compared by each corresponding pair of the actual units.

In sum, Clinical segments exhibited the best performance in ROUGE and it lies between sentences and clauses in their size. Combining the results in this section, we can conclude that the segment units we introduced in this paper are better and optimal units that lie between sentence and clause units.

## 5 Discussion

The result that extractive summarization with sentences is less effective than with other granularities is consistent with previous studies [15, 16]. Given the consistency of these results, this could be a universal property that must be exploited in further summarization tasks in NLP research.

In the summarization of medical documents, the experimental results of using linguistic units suggest that physicians create discharge summaries by capturing clinical concepts from the inpatient records. On the other hand, sentences and clauses performed poorly, probably because they were chunked only with syntactic information and did not deal with medical concepts. Accordingly, automatic summarization in the medical field requires not only syntactic information but also high-level semantic and pragmatic information related to domain knowledge. Clinical segments are reasonable candidates as atomic units that carry medical information. Therefore, clinical segments can potentially be used to quantify the quality of medical documentation and to acquire more detailed medical knowledge expressed in texts.

Limitations in the current study and analysis are twofold: language and cultural dependency. Firstly, Japanese grammar and Japanese medical practices are very different from those of European languages, and there can be differences in the description, summarization, and evaluation processes. Accordingly, this pipeline using extractive method might be applicable only to Japanese clinical setting. In particular, the clinical segment was defined for Japanese, only labeled corpus for Japanese exists, so it is not naively applicable to other languages. However, the idea of capturing medical concepts may be useful for other languages. Also, more

**Table 8. The Relationships between clauses and clinical segments.**

| Relation types | Equal | Inclusive | Included | Overlap |
|---|---|---|---|---|
| Number of relationships | 6,687,046 | 2,239,839 | 1,469,423 | 821,663 |
| | (59.6%) | (20.0%) | (13.1%) | (7.3%) |

researches at various institutions would be preferable to confirm the generalizability of our results, although our study used the largest multi-institutional health records archive in Japan. Secondly, in some countries with different cultural background, *dictation* is used in clinical records and their summaries [63]. In this regard, Japanese hospitals do not use dictation to produce discharge summaries, which could result in frequent copying and pasting from sources to summaries. This custom could have contributed to using extractive texts in the discharge summaries in Japan. The analysis of the influence of this customary difference is left for future work.

## 6 Conclusion

In this study, we explored the best granularity for the automatic summarization of medical documents. The result indicated clinically motivated semantic units, larger than clauses, are the best granularity for the extractive summarization.

Ohter contributions of this study are summarized as follows. First, we defined clinical segments that captured clinical concepts and showed that they can be reliably split automatically by a machine learning-based method. Second, we identified the optimal granularity of extractive summarization that can be used for automated summarization of medical documents. Third, we built a Japanese parallel corpus of medical records with inpatient data and discharge summaries.

The results of this study suggest that the clinical segments that we have introduced are useful for automated summarization in the medical domain. This provides an important insight into how physicians write discharge summaries. Previous studies have used other entities to analyze medical documents [64–66]. Our results will help to provide more effective assistance in the writing process and automated acquisition of clinical knowledge.

## Acknowledgments

The authors would like to thank Dr. Yoshinobu Kano and Dr. Mizuki Morita for their cooperation in our previous research that served as the foundation for this study. We also thank Ms. Mai Tagusari, Ms. Nobuko Nakagomi, and Dr. Hiroko Miyamoto, who served as annotators.

## Author Contributions

**Conceptualization:** Kenichiro Ando, Takashi Okumura.

**Data curation:** Kenichiro Ando, Takashi Okumura, Hiromasa Horiguchi.

**Formal analysis:** Kenichiro Ando.

**Funding acquisition:** Yuji Matsumoto.

**Investigation:** Kenichiro Ando, Takashi Okumura, Hiromasa Horiguchi.

**Methodology:** Kenichiro Ando.

**Project administration:** Takashi Okumura, Yuji Matsumoto.

**Resources:** Takashi Okumura, Mamoru Komachi, Hiromasa Horiguchi.

**Software:** Kenichiro Ando.

**Supervision:** Mamoru Komachi, Hiromasa Horiguchi, Yuji Matsumoto.

**Validation:** Kenichiro Ando.

**Visualization:** Kenichiro Ando.

Writing – **original draft:** Kenichiro Ando, Takashi Okumura.

Writing – **review & editing:** Takashi Okumura, Mamoru Komachi, Yuji Matsumoto.

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
