## [Decision Letter · Decision Letter 0]

24 May 2022

PDIG-D-21-00099

Exploring Optimal Granularity for Extractive Summarization of Unstructured Health Records: Analysis of the Largest Multi-Institutional Archive of Health Records in Japan

PLOS Digital Health

Dear Dr. Okumura,

Thank you for submitting your manuscript to PLOS Digital Health. After careful consideration, we feel that it has merit but does not fully meet PLOS Digital Health's publication criteria as it currently stands. Therefore, we invite you to submit a revised version of the manuscript that addresses the points raised during the review process.

We look forward to receiving your revised manuscript.

Kind regards,

Tom J. Pollard, Ph.D.

Academic Editor

PLOS Digital Health

Journal Requirements:

State what role the funders took in the study. If the funders had no role in your study, please state: “The funders had no role in study design, data collection and analysis, decision to publish, or preparation of the manuscript.”

2. Please update your Competing Interests statement. If you have no competing interests to declare, please state: “The authors have declared that no competing interests exist.”

3. In the online submission form, you indicated that “Our corpus, created in this article, is available upon request. The NHO data is not publicly available for privacy reason.”. All PLOS journals now require all data underlying the findings described in their manuscript to be freely available to other researchers, either 1. In a public repository, 2. Within the manuscript itself, or 3. Uploaded as supplementary information.

4. Please provide separate figure files in .tif or .eps format only and remove any ensure that all files are under our size limit of 10MB.

For more information about how to convert your figure files please see our guidelines: https://journals.plos.org/digitalhealth/s/figures

5. Please ensure that you provide a single, cohesive .tex source file for your LaTeX revision. You may upload this file as the item type 'LaTeX Source File.' As stated in the PLOS template, your references should be included in your .tex file (not submitted separately as .bib or .bbl). Please also ensure that you are making any formatting changes to both your .tex file and the PDF of your manuscript. If you have any questions, please contact Latex@plos.org. You can find our LaTeX guidelines here: https://journals.plos.org/digitalhealth/s/latex

Additional Editor Comments (if provided):

The study is interesting and explores an important topic. Our reviewers - in particular Reviewer 1 - have raised some points that I think should be addressed prior publication. I would be grateful if you could submit a new version of the paper after responding to these points.

Reviewers' comments:

Reviewer's Responses to Questions

**Comments to the Author**

1. Does this manuscript meet PLOS Digital Health’s publication criteria? Is the manuscript technically sound, and do the data support the conclusions? The manuscript must describe methodologically and ethically rigorous research with conclusions that are appropriately drawn based on the data presented.

Reviewer #1: Yes

Reviewer #2: Yes

2. Has the statistical analysis been performed appropriately and rigorously?

Reviewer #1: Yes

Reviewer #2: I don't know

3. Have the authors made all data underlying the findings in their manuscript fully available (please refer to the Data Availability Statement at the start of the manuscript PDF file)?

Reviewer #1: Yes

Reviewer #2: No

4. Is the manuscript presented in an intelligible fashion and written in standard English?

Reviewer #1: Yes

Reviewer #2: Yes

5. Review Comments to the Author

Reviewer #1: An insightful presentation of the state of the art for clinical health records and discharge summaries in Japanese hospitals as well as the application of natural language processing (NLP) on said clinical text data. NLP in Japanese, let alone for clinical discharge summaries, presents a unique challenge and we applaud the authors for gaining access to the largest clinical text database and the efforts for conducting this study.

However, there were a few key elements that the paper could be revised in order to showcase and highlight the novelty of the study. Suggestions are not limited to but include:

● The paper did not describe the unique difficulties in conducting NLP in Japanese, such as the lack of spaces between the characters and words. This is further exacerbated with the lack of open source information to combat this, and including such limitations may further emphasize the quality of this study.

● The authors do not declare any other NLP related models or research available in Japan, and the study does not list many related studies such as the following which would have been helpful in creating comparisons:

→ Preliminary development of a deep learning-based automated primary headache diagnosis model using Japanese natural language processing of medical questionnaire: https://www.ncbi.nlm.nih.gov/pmc/articles/PMC7827501/

→ Predicting Inpatient Falls Using Natural Language Processing of Nursing Records Obtained From Japanese Electronic Medical Records: Case-Control Study: https://medinform.jmir.org/2020/4/e16970/

→ A clinical specific BERT developed using a huge Japanese clinical text corpus: https://journals.plos.org/plosone/article?id=10.1371/journal.pone.0259763

● Relatedly to the aforementioned point, due to the lack of related studies, the authors do not mention how different or novel their research stands in comparison to other Japanese or oversea models.

● Furthermore, the authors mention that the results might be applicable only to Japanese clinical settings, yet since no other examples are given, this cannot be generalized and further examples as well as background information and explanation is required to make this claim.

● The authors mention that their model requires entire documents for training, and that the number of documents in their corpus was too small to be used for the training of the model; they however do not explain why this is the case. Perhaps by including examples from Japan, if not from abroad, would help readers understand their model’s novelty.

● The authors mention that in this experiment, they used three contextual units instead of n-gram units to evaluate their impact on the summarization performance and determine which unit performs the best. While this is interesting, the authors once again do not describe why they did this and why this is meaningful.

● The authors discuss that this study had used the largest multi-institutional health records archive in Japan and thus it would be worthwhile to validate the results in multiple languages, but if there are actually no other databases, perhaps the authors must first be validating in other Japanese clinical text databases, or other Asian languages that have similar grammatical structures, prior to applying firsthand to other languages that may have more open source resources available.

● The authors raised an interesting point that Japanese hospitals do not use dictation to produce discharge summaries, which could result in frequent copying and pasting from sources to summaries. However, it would be stronger for the authors to provide examples for how this enhances (or does not enhance) the model, as well as further expand on the clinical significance or rationale to dictate or not dictate, and whether there are differences in clinical outcomes or clinical workflow to explore the strengths and weaknesses of the authors’ model.

● Finally (and related to the point above), the authors do not clearly discuss why this study is useful, especially its application to real world clinical settings. 

While the English and Japanese translations available are also interesting, some of the translation seem non-native or slightly incorrect, thus perhaps could use proof-reading and reconsideration (e.g. 髄液糖定量 - glucose determination → cerebrospinal fluid glucose level or glycorrhachia). 

Given all these considerations, priority for PLOS Digital Health will be based on significant revisions, only after being able to convince the editor that their model has significance in real-world clinical settings in comparison to pre-existing models.

Reviewer #2: The study aims to identify the optimal granularity between 3 proposed granularities for performing extractive summary of Japanese clinical texts.

The paper is clear and well-written.

Comments:

1) Taking into consideration that the exploration of optimal granularity for extractive summary had beed previously explored, the innovation of the research, is the choice of the clinical domain. However it is not clear if the results were different from the results reached in other non-clinical domains. 

2) The fact that “20-31% of the sentences in discharge summaries were created by copying and pasting", made the authors conclude that a certain amount of content can be automatically generated by extractive summarization. 

However since most of the summary needs to be generated by other methods (e.g. abstractive summary), the fact that extractive summarization cannot be used to generate a full summary, raises questions about its use at all in the summarization process of clinical summaries. Would be whether other methods would make any use of the 20% that can be generated by extractive summary?

3) The background should be further expanded, for example, there are no mention about abstractive methods that create summaries without halluciantions, the authors just mention abstractive summaries "often produces fake contents that do not match the reference summary". A more complete review of previous studies should also include abstractive methods, such as implemented in the CliniText system and BT-45 for example, that did not produce any hallucinations. 

Minor comments:

1) Typos: "In this study,105we adopt both of the two methods. in106Japanese"

2) 

"However, it remains unclear how the summaries should be generated from the unstructured source"

It’s unclear what the authors meant by unstructured source ? Images ? Free-text from other clinical experts ? Raw data ?

6. PLOS authors have the option to publish the peer review history of their article (what does this mean?). If published, this will include your full peer review and any attached files.

**Do you want your identity to be public for this peer review?** For information about this choice, including consent withdrawal, please see our Privacy Policy.

Reviewer #1: No

Reviewer #2: No

---

## [Editor Report · Decision Letter 1]

28 Jul 2022

Exploring Optimal Granularity for Extractive Summarization of Unstructured Health Records: Analysis of the Largest Multi-Institutional Archive of Health Records in Japan

PDIG-D-21-00099R1

Dear Dr. Okumura,

We are pleased to inform you that your manuscript 'Exploring Optimal Granularity for Extractive Summarization of Unstructured Health Records: Analysis of the Largest Multi-Institutional Archive of Health Records in Japan' has been provisionally accepted for publication in PLOS Digital Health.

Best regards,

Imon Banerjee

Section Editor

PLOS Digital Health

Many thanks for your detailed response to the reviewer comments. I am satisfied that the concerns have been addressed and would be happy to move ahead with publication.